# Domiciliation of *Trichoderma asperellum* Suppresses *Globiosporangium ultimum* and Promotes Pea Growth, Ultrastructure, and Metabolic Features

**DOI:** 10.3390/microorganisms11010198

**Published:** 2023-01-12

**Authors:** Zeiad Moussa, Yasmene F. Alanazi, Aiah Mustafa Khateb, Noha M. Eldadamony, Marwa M. Ismail, WesamEldin I. A. Saber, Doaa Bahaa Eldin Darwish

**Affiliations:** 1Microbial Activity Unit, Microbiology Department, Soils, Water and Environment Research Institute, Agricultural Research Center, Giza 12619, Egypt; 2Department of Biochemistry, Faculty of Science, University of Tabuk, Tabuk 71421, Saudi Arabia; 3Medical Laboratory Technology Department, College of Applied Medical Sciences, Taibah University, Madinah 42353, Saudi Arabia; 4Seed Pathology Research Department, Plant Pathology Research Institute, Agricultural Research Center, Giza 12619, Egypt; 5Central Lab of Biotechnology, Plant Pathology Research Institute, Agricultural Research Center, Giza 12619, Egypt; 6Botany Department, Faculty of Science, Mansoura University, Mansoura 35516, Egypt

**Keywords:** *Globisporangium ultimum*, hydrolytic enzymes, mycoparasitism, GC-MS, electron microscopy, dehydrogenase, microbial community

## Abstract

The beneficial microorganisms represent a new and hopeful solution for a sustainable environment and development. In this investigation, *Trichoderma asperellum* ZNW, isolated from seeds, was domiciliated within the pea plant for improving growth, disease management, and enhancement of productivity. *Globisporangium ultimum* NZW was isolated from deformed pea seeds, representing the first record of the pathogen caused by pea damping-off. Both fungi were molecularly identified. *T. asperellum* ZNW produced several lytic enzymes and bioactive metabolites as detected by GC-MC. The SEM illustrated the mycoparasitic behavior of *T. asperellum* ZNW on *G. ultimum* NZW mycelia. In the pot experiment, *T. asperellum* domiciliated the root and grew as an endophytic fungus, leading to root vessel lignification. Under soil infection, *T. asperellum* reduced damping-off, by enhancing peroxidase, polyphenol, total phenols, and photosynthetic pigments content. The vegetative growth, yield, and soil dehydrogenase activity were improved, with an enhancement in the numerical diversity of the microbial rhizosphere. This work may enable more understanding of the plant-fungal interaction, yet, working on domiciliation is recommended as a new approach to plant protection and growth promotion under various ecological setups.

## 1. Introduction

The current era shows great awareness of environmental problems. Microbial diversity can help in sustaining a clean and balanced ecosystem. Plant-associated microorganisms, especially endophytes, represent a hopeful tool for ecological maintenance and sustainable development. To do so, beneficial microorganisms have several mechanisms, e.g., acting as biofertilizers, bio-stimulants, and increasing resistance against abiotic and biotic stresses. Economically, applying plant-associated microorganisms is expected to drop production budgets [1].

Besides the ecological role, plant growth-promoting fungi (PGPF) is useful fungi that have dual uses; they stimulate the growth of plants and protect plants from pathogens. The group of PGPF contains diverse species of fungi, e.g., *Trichoderma* spp., non-pathogenic species of *Aspergillus,* and *Penicillium*, however, more investigations to elucidate the biological role of PGPF are encouraged [2].

*Trichoderma* spp. contains diverse fungi that are present in different environments, mostly in soils. *Trichoderma* spp. can be linked with plants such as epiphytes, endophytes, and/or in the rhizosphere region, acting as promotors of plant growth and/or bioagents against various plant diseases. *Trichoderma* species secrete diverse volatile and nonvolatile compounds that display antibiotic activity [3]. Additionally, *Trichoderma* spp. produces many compounds that improve the plant’s defense, growth, and resistance to numerous biotic and abiotic stresses like heavy metals, salinity drought, cold, etc. Several biocontrol modes were reported by *Trichoderma* spp., i.e., (i) the production of hydrolytic enzymes, antibiotics, and many bioactive metabolites, (ii) competition with the pathogen for nutrients, and space, (iii) mycoparasitism, and/or (iv) induction of defense-related genes and systemic resistance in plants [4].

The bioactive metabolites, especially lytic enzymes squirted by *Trichoderma* spp., hinder the growth of plants’ pathogens. For example, proteases, glucanases, and chitinases have been stated to hydrolyze the cell wall of the pathogen. Also, extracellular proteins produced by *Trichoderma* spp. have a vital task in the inhibition of phytopathogens, as well as the enhancement of plants’ immune response [5]. These modes of action may indirectly or directly suppress the phytopathogens’ growth and pathogenicity.

Furthermore, as bio-stimulants, *Trichoderma* spp. improves plant health and productivity through the production of peptides, hormones, volatiles, and nonvolatile metabolites that enhance the growth, and rooting system [6]. *Trichoderma* spp. induces the plant to produce defense-related enzymes, such as polyphenol oxidase and peroxidase, which improve defense response against fungal, viral, and bacterial plant pathogens [7,8,9]. Ecologically, the presence of *Trichoderma* spp. in soil improves soil microbial communities leading to improvement of plant growth and yield [10].

Pea (*Pisum sativum* L.) is grown in numerous regions in the world and is consumed by animals and humans. It is known as a low-cost, readily available source of protein, minerals, vitamins, and complex carbohydrates [11]. The world cultivated area of pea reached 2.5 million ha, yielding 19.8 million tons [12]. Pea is exposed to the attack by several fungal phytopathogens, causing significant decreases in quality and yield. For instance, the damping-off disease, caused by different abiotic or biotic stresses, leads to an obvious decline in seed germination and seedling emergence, or development. Plant pathogenic fungi have been stated as the most vital biotic stress. *Fusarium* spp., *Pythium* spp., *Phytophthora* spp., and *Rhizoctonia* spp. are the greatest commonly damping-off pathogens [13]. Thus, this disease is one of the most significant yield constraints in fields and nurseries.

To the authors’ information, the existing pioneer work is the first state of domiciliation of a new *Trichoderma* sp., isolated from pea seeds, into pea plant tissue as a new habitat for the bioagent fungus. The growth and health-promoting action, as well as the biocontrol of pea damping-off disease, were investigated. Evidence of the domiciliation process of the endophytic fungus was followed up by ultrastructural and biochemical studies. Additional investigation on the changes in the rhizosphere microbial biodiversity was carried out.

## 2. Materials and Methods

### 2.1. Isolation Trial

Two types of samples of pea seeds (Master B) were collected. The first were deformed seeds obtained during a routine work of seed investigation. This sample was used for the isolation of the causal pathogenic fungus. The second was healthy-looking seeds and was used for the isolation of the bioagent. The seed-borne fungi were isolated by the standard blotter method. About 100 seeds for each seed type were separately surface sterilized by sodium hypochlorite (5 min), then washed away with sterilized water (10 min), and dried with filter paper under aseptic conditions. Ten seeds of each type were plated before being incubated at 25 ± 2 °C with daily follow-up until the appearance of fungi, which were isolated, purified, and kept on PDA slants until testing [14]. The common phytopathogenic fungus that repeatedly appeared on the deformed seeds was isolated and further subjected to the pathogenicity test on pea seeds.

### 2.2. Pathogenicity Test

To confirm the severity of the pathogenic fungus, the pathogenicity test was performed. The inoculum of the pathogenic fungus was prepared by growing a 5 mm fungal disk (obtained from the edge of a 5-day-old colony) on PDA plates. After incubation, the spores’ suspension was prepared by scrapping the fungal spores into 50 mL of sterilized distilled water. For the pathogenicity test, surface-sterilized (using 1% sodium hypochlorite) healthy seeds of pea were immersed in each spore suspension of the tested fungus, containing Arabic gum solution (2%, for 15 min), being used as an adhesive agent, the seeds were left for drying at room temperature. The control treatment was prepared by immersing healthy seeds in distilled water. Under greenhouse conditions, 10 seeds per pot were planted. The most aggressive fungal isolate was selected for additional study.

### 2.3. Antagonism Test

The dual culturing method was used to evaluate the efficiency of the selected *Trichoderma* spp. isolate against the damping-off causal fungus [15]. A 5-mm mycelial disc of the tested bioagent was paired against a mycelial disc of the fungal pathogen at the opposite side of a 9-cm diameter PDA plate. Both the discs of the antagonist and the pathogen were inoculated at 1 cm apart from the plate edge, then incubated at 25 °C. The radial growth of the control and the pathogen were measured. The inhibition in radial growth of the pathogen was calculated.

To estimate the antifungal activity of the volatile metabolites of *Trichoderma* spp., two bottoms of Petri dishes with PDA were separately inoculated with either a disc of *Trichoderma* spp. or a disc of the tested fungal phytopathogen. The two bottoms were adjusted and connected by tape. The control treatment was prepared without the pathogen only [16].

The mycoparasitic behavior was inspected using scanning electron microscopy (SEM). After the first contact between *Trichoderma* spp. and phytopathogen in a dual culture, the mycelial interaction zone (0.5 cm) was cut from the interface region. The sample was vapor-fixed for 20 h with osmium tetroxide (2%, *w*/*v*), dried, and held in a desiccator [16,17], then mounted on copper, before being coated with gold-palladium (sputter coater, JFC-1100). The examination was done at different magnifications by SEM (TEM-2100, JEOL, Tokyo, Japan) connected to an accelerating voltage unit.

### 2.4. Morphological, Microscopic, and Molecular Identification

The isolated fungi were subjected to cultural, light microscopic, and scanning electron microscopy identifications [18,19], followed by molecular identification and evolutionary relationship.

For molecular identification, the internal transcribed spacer (ITS) analysis was performed using a polymerase chain reaction (PCR). The reaction of the PCR amplification was achieved in 50 µL composed of 1.5 mM MgCl_2,_ 2.5 mM dNTPs, 1X reaction buffer, 1U *Taq* DNA polymerase (Promega, Madison, WI, USA), and 30 picomole of each primer (ITS4 R; 5′-TCCTCCGCTTATTGATATGC-3′ and ITS1 F; 5′-TCCGTAGGTGAACCTGCGG-3′) with a product size of 600 bp, and 30 ng DNA genome.

The thermo-cycling PCR technique was applied for the amplification process (Perkin-Elmer/GeneAmp^®^ PCR System 9700, PE Applied Biosystems, Waltham, MA, USA) through fulfilling forty cycles after a cycle of initial denaturation at 94 °C for five min. Each cycle is composed of the following steps; thirty-sec denaturation (94 °C), thirty-sec annealing (45 °C), and one min elongation (72 °C). The final cycle of the primer extension segment continued for seven min (72 °C). The PCR outcomes were resolved by agarose gel electrophoresis (1.5%) and ethidium bromide (0.5 µg/mL) at 95 volts in 1X TBE buffer. A molecular size standard was used (100 bp DNA ladder). UV light was utilized for visualizing PCR products and a BIO-RAD-2000 Gel Documentation System was used for photographing these products.

The amplified and purified PCR products were performed by mixing the PCR mixture into a microfuge tube (1.5 mL) with three volumes of binding buffer, then transferring it to the EZ-10 spin column, left at room temperature for two min, and centrifuged. Next, it was mixed with 750 µL of wash solution and centrifuged (10,000 rpm, 2 min); the procedure was repeated to remove any residuals, followed by transferring it to a microfuge tube (1.5 mL) with elution buffer (50 µL).

The analysis of ITS sequencing of the PCR product was achieved (Automatic sequencer ABI PRISM 3730XL Analyzer, Thermo Fisher Scientific, Cincinnati, OH, USA) by utilizing sequencing kits (Big Dye TM Terminator Cycle). Using the Rbcl Forward primer, single-pass sequencing was accomplished on every template. The product was purified and exposed to electrophoresis (ABI 3730xl sequencer, Microgen Company, Moscow, Russia). The alignment was performed (Align Sequences Nucleotide BLASTn (http://www.ncbi.nlm.nih.gov/BLAST, accessed on 31 May 2021). The relationships of taxa were compared based on the Neighbor-Joining method [20]. The replicate trees’ percentage (1000 replicates) was compared [21]. The distance of the evolution was calculated for each sequence pair, using the pairwise deletion option [22]. The ambiguous positions were eliminated. MEGA11 was used for evolutionary analyses [23].

### 2.5. Biochemical Features of Trichoderma sp.

#### 2.5.1. Assay of Lytic Enzymes

The lytic capacity of *Trichoderma* spp. was evaluated. The inoculum was prepared by growing the isolate on Czapek Dox agar plates. Two 0.5 mm disks of 7 days-old culture were used to inject the fermentation medium. The culturing technique was done using dried crushed pea plant (1.0 g) as a solid-state fermentation (SSF) medium after being humidified with 5 mL tap water in 250 mL Erlenmeyer flasks. The flasks were sterilized, inoculated (15 min, 121 °C), and incubated (30 °C, for 7 days). Distilled water (10 mL) was added to the grown culture and shaken (30 min, 150 rpm), then centrifuged (5000 rpm, 20 min). The resultant filtrate was assessed for lytic enzymes.

Xylanase enzyme was assessed in the fungal filtrate by using xylan as substrate [24], while for cellulase, microcrystalline cellulose was used [25]. Polygalacturonase (pectinase) was also assessed in a mixture of pectin and filtrate [26]. The reducing units, due to the enzymatic actions of the three enzymes, were determined at A_575_ nm [27]. The enzyme unit (U) was the amount that liberates one µmol/g/min of xylose (xylanase), glucose (cellulase), or D-galacturonic acid (pectinase) under the assay.

Proteinase enzyme was measured in casein as a substrate. The free amino acids were determined at A_280_ [28]. One protease U was expressed as the quantity that resulted in the liberation of one µg equivalent of tyrosine/g/min.

Chitin azure was used to measure chitinase at A_575_ nm. The U of chitinase was expressed as the amount that increases the absorbance by 0.01 [29].

#### 2.5.2. Gas Chromatography-Mass Spectrometry Analysis (GC-MS)

For derivatizations of metabolites, *Trichoderma* spp. was grown in potato dextrose broth after incubation (150 rpm and 25 °C for 7 days; then, the grown fungal culture was centrifuged at 10,000 rpm for 20 min). The filtrate was resuspended (50 µL of BSTFA incubated in a Dry Block Heater) at 70 °C for 30 min. The GC-MS (Agilent Technologies, Santa Clara, CA, USA) was supported with a mass spectrometer detector (5977A) and a gas chromatograph (7890B). The GC was supported with columns (HP-5MS, 30 m × 0.25 mm diameter and 0.25 μm film width). The carrier gas (hydrogen) was added at 1.0 mL/min, with initial thermo-programming (50 °C for 1 min), which rose 10 °C/min before holding (20 min) at 300 °C. The injector and detector were held at 250 °C. The mass spectra were obtained by electron ionization at 70 eV; by a spectral range of 30–700 *m/z* and a solvent delay of 9 min. The mass temperature was 230 °C and Quad 150 °C. The different ingredients were compared with the spectrum patterns (Wiley and NIST Mass Spectral Library data).

### 2.6. Greenhouse Experiment

#### 2.6.1. Pea Seeds and Fungal Inocula

The master B cultivar of pea seeds was obtained from the Research Institute of Horticulture, Agricultural Research Center, Giza, Egypt.

The inoculum of the damping-off pathogen (*Globisporangium ultimum* NZW) was on plates of PDA after incubation for five days (at 23 °C), then mycelial plugs were transferred to sterilizing medium of sorghum: water: coarse sand: (2:2:1 *v*/*v*) and incubated (23 °C for 10 days).

The inoculum of the bioagent (*Trichoderma asperellum* ZNW) was prepared in potato dextrose broth under static conditions (7 days at 25 °C incubation temperature).

#### 2.6.2. Setting-Up the Pot Experiment

The experiment was performed to study the proficiency of *Trichoderma asperellum* (the bioagent) to enhance the health and the growth of pea and to antagonize the damping-off phytopathogen, *Globisporangium ultimum*. The experiment was carried out from 17 December 2020 to 17 February 2021, at TagElezz Agricultural Research Station, Dakahlia, Egypt (Lat. 030° 57′ 25″ N, Long. 031°35′ 54″ E).

Natural soil (sand: clay, 1:2 *v/v*) was used for filling pots (8 kg/pot). Half the number of the pots was singly inoculated with *G. ultimum* NZW (0.4%, *w*/*w*), irrigated, and left for 7 days to confirm the spreading of the fungus into the soil. While the other half of the pots were non-infested by the pathogen. At the time of planting, the pea seeds were soaked in the filtrate of *T*. *asperellum* ZNW for one hour, while others were soaked in fungicide. Chemical fungicide (Unidron, SC 56%) was used at the recommended dose that recommended by the Egyptian Ministry of Agriculture (250 cm/100 L).

Accordingly, the applied treatments were; (1) fungicide, (2) control (without any treatments), (3) *T*. *asperellum* ZNW (T), (4) Fungicide + *G. ultimum* NZW (5), *G. ultimum* NZW, and (6) *T*. *asperellum* ZNW + *G. ultimum* NZW.

#### 2.6.3. Disease Assessment, Vegetative Growth, and Yield

After 15 days of planting, the rotted pea percentage (un-emerged peas) and post-infected seedlings were recorded. The plant survival after 30 days from planting was estimated.

The vegetative growth parameters were measured for each treatment. Three plants were neatly harvested after 5 weeks from planting and rinsed with tap water to eliminate the soil’s particles. Shoot, root, and plant lengths (cm), the fresh and dry weight of root (g), the fresh and dry weight of plant (g), and leaf area (cm^2^) were measured.

Three months after planting, the yield parameters were determined in terms of the number of pods, the fresh and dry weight of pods, and the 100 seeds’ weight.

#### 2.6.4. Root Anatomy

Changes in the anatomy in pea root due to the phytopathogen and bioagent treatments were studied in the root cross sections, after gold-coating, and examined by SEM (TEM-2100, JEOL, Tokyo, Japan) connected to an accelerating voltage of 30 kV [30].

#### 2.6.5. Physiology Features

The biochemical features of pea plants under different treatments were measured after 45 days of planting. Total phenolic content was estimated spectrophotometrically at 650 nm [31]. The activities of peroxidase and polyphenol oxidase were measured spectrophotometrically at 470 and 495 nm, respectively. The enzyme unit was expressed as U min/g plant fresh weight [32].

#### 2.6.6. Chlorophylls and Carotenoids Content

Chlorophylls (a, b, and total), and carotenoids were determined. A 0.05 g sample of fresh leaves taken from the first plant leaflet was soaked overnight in 10 mL of methanol with a trace of sodium bicarbonate (4 °C) to hinder chlorophyllase enzyme activity [33]. The chlorophyll contents were then assessed spectrophotometrically at 452.5, 650, and 665 nm, then chlorophylls and carotenoids were estimated [34].

### 2.7. Biological Features of Soil

#### 2.7.1. Soil Dehydrogenase Activity

After 60 days of planting, dehydrogenase activity was determined in the rhizosphere soil [35]. One gram of sieved soil was mixed with 1 mL of 3% aqueous solution (*w*/*v*) 2,3,5-triphenyl tetrazolium chloride (TTC) and stirred well with a glass rod. After the incubation (96 h and 27 °C), 10 mL of ethanol was added to each test tube, then vortexed (30 s), and incubated for 1 h to permit the settling of the suspended soil. Five ml of the resultant filtrate was transferred to a test tube to measure dehydrogenases that lead to the conversion of TTC to triphenyl-formazan (TPF). The formazan’s absorbance was recorded spectrophotometrically at A_485_ nm. The unit of dehydrogenase activity is expressed as µg TPFg^−1^ dry soil min^−1^.

#### 2.7.2. Soil Microflora

After 60 and 90 (the end of the experiment) days, the rhizosphere soil was gathered, and the plants’ debris was removed. Bacterial and fungal counts were determined [36]. Ten grams of soil were shaken (120 rpm/min for 30 min) with sterile water (90 mL), then serial dilution was prepared. Plates of nutrient agar medium were used for counting the total bacteria after incubation (3 days at 28 °C). The PDA medium was used for the estimation of total fungal count after incubation (25 °C for 7 days). The number of colonies was determined and the log value of CFU g^−1^ dry soil was calculated.

### 2.8. Statistical Analysis

At least all trials were performed in triplicates. The greenhouse trial was arranged in a completely randomized block. The ANOVA was performed, followed by means comparison (Tukey’s HSD) at a probability (*p* ≤ 0.05). The CoStat software package (version 6.450, CoHort Software, Monterey, CA, USA) was used.

## 3. Results

### 3.1. Fungal Isolates

*Trichoderma asperellum* ZNW and the pathogen (*G. ultimum* NZW) were isolated from the healthy and deformed pea seeds, respectively. The two fungi were grown on the PDA plate. *Trichoderma asperellum* had white mycelia with a cottony texture and green spores. The mycelia of the most aggressive damping-off phytopathogenic fungus had a white color.

### 3.2. Antagonism Test

The antagonism between the bioagent and pathogen was tested directly on the agar plate. *Trichoderma asperellum* ZNW caused a reduction of the growth fungal pathogen; the reduction increased with the incubation time, recording 43.01 ± 3.02, 72.17 ± 2.05, and 82.25 ± 1.10% after 24, 72, and 120 h, respectively. *Trichoderma asperellum* ZNW could overgrow *G. ultimum* NZW, the causative agent of the damping-off of the pea (Figure 1).

Another antagonistic activity test based on the volatile metabolites of *Trichoderma* spp. ZNW was performed. There is no observed inhibition activity of the volatile compounds of *T. asperellum* against the fungal pathogen where there was no difference between the fungal pathogen’s growth on the control and the dual culture plates.

### 3.3. Morphology and Antagonism Investigation by SEM

The morphology of the two fungi was further examined by SEM. The conidia of *T. asperellum* are oval or slightly globose, measuring 3.75 um in length and 2.5 um in width. The branched conidiophores measure 1.5 um in width. Every five conidia are arranged together to resemble a flower (Figure 2A).

The diameter of the hyphae of the pathogenic fungus has about 2.5 um diameter with terminal hyphal swellings (Figure 2B). Based on cultural, light microscopy, and SEM investigations, both fungi were identified as *Trichoderma asperellum* ZNW, and *Globisporangium ultimum* NZW.

The images of mycoparasitism of *T. asperellum* ZNW on the pathogenic fungus; *G. ultimum* NZW under SEM showed the coiling of *T. asperellum* ZNW hyphae around *G. ultimum* NZW (host) and caused the breakdown and collapse of the host hyphae (Figure 3) at different stages of mycoparasite. As shown, *T. asperellum* ZNW overgrows *G. ultimum* NZW (Figure 3A), then *T. asperellum* ZNW hyphae coiled around *G. ultimum* NZW hyphae (Figure 3B,C), leading to the breakdown and collapse of *G. ultimum* NZW hyphae due to the pressure of *T. asperellum* ZNW hyphae (Figure 3D,E).

### 3.4. Molecular Identification

The fungal isolates were identified molecularly, after PCR amplification, through ITS sequencing, which codes for the 18S rRNA gene. The gel electrophoresis showed that the amplified ITS fragment (600 bp) of both fungi was similar to the target sequence of the related fungal. The phylogeny was nonstructured (Figure 4A,B). The nucleotide sequences and the BLAST assessment show that the bioagent strain (ZNW) exhibited a high similarity (99%) with the *Trichoderma asperellum* on the GenBank (Figure 5A). The fungus was identified as *Trichoderma asperellum*, and the GenBank accession number is OL604172. The pathogenic fungal isolate NZW showed similarity with more than 95% with the closely related *Globisporangium ultimum* on the GenBank (Figure 4B). The strain NZW was identified and classified as *Globisporangium ultimum* NZW, with accession number MW922713.

### 3.5. Profile of Lytic Features

The prototype of the hydrolytic enzymes of the bioagent fungus was assayed (Figure 5) as a possible biocontrol method. The fungus produced all tested enzymes, including xylanase, cellulase, pectinase, protease, and chitinase. *Trichoderma asperellum* ZNW in general demonstrated a high amount of all tested enzymes at various levels.

### 3.6. GC-MS Assessment

The *T*. *asperellum* ZNW metabolite was investigated. Results of the GC-MS analysis (Table 1 and Figure 6) indicated that the main components of *T*. *asperellum* ZNW filtrate were 9-octadecenoic acid (Z) (peak area-36.32%), Oleamide, TMS derivative (peak area-29.89%),1,2-15,16-diepoxyhexadecane (peak area-8.13%), 1-Monopalmitin, 2TMS derivative (peak area-5.74%), and 9-hexadecenoic acid (palmitic acid) (peak area-3.96%). Whereas, other components were present in low percentages; acetic acid, (1,2,3,4,5,6,7,8-octahydro-3,8,8-trimethylnaphth-2-yl), methyl ester (peak area-2.5%), 9,12-octadecadienoyl chloride (1.81%), 1,4-Bis(trimethylsilyl)benzene (peak area-1.36%), and butylated hydroxytoluene (peak area-1.16%). On the other hand, there are many components present in traces (peak area less than 1%).

### 3.7. Greenhouse Experiment

#### 3.7.1. Disease Development

The bioagent was further evaluated under greenhouse conditions as a possible biocontrol for damping-off disease. Infection by *G. ultimum* NZW (infested soil only) recorded the highest values of rotted seeds and infected seedlings, being 46.67% and 18.33%, respectively, and the lowest number of plants that survived, 35% (Table 2). On the other hand, there were no significant differences among the other treatments, where priming pea seeds by fungicide or *T. asperellum* ZNW increased the percentage of survival and decreased the percentage of rotted seeds and infected seedlings in the presence or absence of *G. ultimum* NZW. Generally, the highest survival percentages were recorded in both fungicide (80%) and *T. asperellum* ZNW (80%) treatments, followed by *T. asperellum* + *G. ultimum* (78.33%). Whereas the survival percentage of the control and fungicide + *G. ultimum* treatments was 70% each. Finally, *T. asperellum* ZNW alone led to an increase in plant survival by 10 compared to non-infected control, and by 45% under the soil infestation with *G. ultimum*, in addition to the decrease in the rotted seeds and infected seedlings.

#### 3.7.2. Vegetative Growth Parameters

All treatments by *T. asperellum* ZNW or fungicide had higher vegetative growth parameters than those non-treated ones. *Trichoderma asperellum* ZNW positively improved plant growth (root, shoot, and plant lengths, as well as the fresh and dry weight of root, the fresh and dry weight of the plant, and leaf area) (Table 3). Moreover, under infection, *T. asperellum* ZNW improved all previous vegetative characteristics as compared with pathogen-only treatment. The highest recorded value of root length was 10 cm, recorded by *T. asperellum* ZNW + *G. ultimum* NZW treatment, while the lowest root length (5.33 cm) was observed with the treatment of infested soil by *G. ultimum* NZW. Also, the plants treated with fungicide improved (significantly or insignificantly) the root and plant weight (fresh and dry) and leaf area more than the untreated ones.

#### 3.7.3. Ultrastructural Changes of Pea Root

The cross-section magnified by SEM showed normal control plants (Figure 7A), and also showed the ability of *T. asperellum* hyphae to domiciliate and grow as an endophytic fungus in pea roots (Figure 7B). Moreover, *T. asperellum* ZNW led to thickening in the cell wall of vessels (xylem). The cross-section of pea plants that were planted in the infested soil by *G. ultimum* NZW showed an accumulation of *G. ultimum* NZW spores in plant vessels, leading to damping-off (Figure 7C). Moreover, the cross-section of pea root treated by *T. asperellum* ZNW and planted in soil infested by *G. ultimum* NZW showed the growth of *T. asperellum* ZNW and thickening in the cell wall of vessels (the lower part of the photo) that prevents *G. ultimum* NZW spores (that accumulate in the upper part of the photo) to invade the region of root in which *T. asperellum* ZNW present (Figure 7D).

#### 3.7.4. Defense Response

The total phenols, peroxidase, and polyphenol oxidase of different treatments, that were planted in non-infested and infested soil with *G. ultimum* NZW, are illustrated in Table 4. Generally, the plants treated with *T. asperellum* ZNW and fungicide showed enhancement in the accumulation of total phenols and the obvious activity of both defense-related enzymes. Under infection, *T*. *asperellum* ZNW also significantly enhanced the total phenol content and the activity of defense-related enzymes as compared with *G. ultimum* NZW treatment. The highest peroxidase activity was observed with the treatment of *T. asperellum* ZNW + *G. ultimum* NZW (14 U). Also, no significant variation was observed in the value of peroxidase activity of the two treatments, the fungicide, and the control. *G. ultimum* NZW treatment has the lowest value of peroxidase activity (4.7 U), and polyphenol oxidase activity (4 U). Other treatments had a significant increase in polyphenol oxidase activity where there was no significant variation between *T. asperellum* ZNW + *G. ultimum* NZW, *T. asperellum* ZNW, fungicide, and the control, being 11.3, 11.3, 10, and 8 U, respectively. Moreover, plants treated with *G. ultimum* NZW had the lowest value of total phenols (43 mg), whereas those treated with *T. asperellum* ZNW + *G. ultimum* NZW had the highest value of total phenols (71.0 mg).

#### 3.7.5. Photosynthetic Pigmentation

Chlorophylls (a, b, and total), and carotenoids were determined as a sign of metabolic changes (Table 5). Under infested soil, *G. ultimum* NZW showed a variable significant number of photosynthetic pigments in pea leaves when compared with other treatments. Under infection stress, *T*. *asperellum* ZNW showed a significant variation in total chlorophyll compared to the other treatments. The treatment of *T. asperellum* ZNW + *G. ultimum* NZW had the highest significant values of chlorophyll a (2.71 mg g^−1^ fresh weight), chlorophyll b (0.987 mg g^−1^ fresh weight), and total chlorophylls (3.694 mg g^−1^ fresh weight). Whereas no significant alterations were noticed among values of carotenoids of different treatments.

#### 3.7.6. Yield Parameters

Finally, the recorded yield parameter of pea plants treated by *T. asperellum* ZNW or fungicide were significantly increased than the corresponding values of untreated plants (Table 6). The treatment by *T. asperellum* ZNW planted in infested soil (*T. asperellum* ZNW + *G. ultimum* NZW) had the highest values of yield parameters (70.67 pods, 81.07 g fresh weight of pods, 14.28 g pods dry weight, and 8.63 g weight of 100 seeds). On the contrary, *G. ultimum* NZW had the lowest values of yield parameters (35 pod of 28.65 and 5.05 g of fresh and dry weights, respectively, and 6.23 g of 100 seeds).

### 3.8. Biological Activity of the Rhizosphere Soil

#### 3.8.1. Dehydrogenase Activity

The treatment of fungicide and *T. asperellum* + *G. ultimum* led to enhancements of dehydrogenase activity in the rhizosphere soil (0.283 and 0.404 µg TPF g^−1^ dry soil min^−1^, respectively) when compared with the other treatments (Table 7). The treatment of infested soil only had the lowest recorded value (0.211 µg TPF g^−1^ dry soil min^−1^).

#### 3.8.2. Population of Soil Microflora

The total bacterial and fungal counts in the rhizosphere soil after 60 days of planting were more than the corresponding values after 90 days of planting (Table 8). After 60 days of planting, the highest recorded total bacterial count was observed with the infested soil by *G. ultimum* NZW (log 6.99), while the value of the total bacterial count of the treatments of *T. asperellum* ZNW and fungicide was lower than the corresponding value in the infested or non-infested soil.

After 90 days of planting, the highest total bacterial count was observed with the control treatment (log 6.29), whereas the lowest value was reported with infested soil with *G. ultimum* NZW (log 5.64). However, the total fungal count in *T. asperellum* and fungicide treatments in the non-infested soil was lower than that of the control treatment.

The highest count of soil fungal count, after 60 days, was recorded with the treatment by *T. asperellum* ZNW (log 4.99) while the lowest value was reported with the treatment *T. asperellum* ZNW + *G. ultimum* NZW (log 4.46). The fungal count of plants treated with *T. asperellum* ZNW or fungicide was higher than the corresponding value of the control and *T. asperellum* + *G. ultimum* treatments. After 90 days, the control and fungicide + *G. ultimum* treatments, followed by *T. asperellum* ZNW, recorded the highest soil fungal counts.

## 4. Discussion

Owing to their biodiversity in various ecosystems, beneficial fungi must be considered when evolving plant protection strategies. Unfortunately, public awareness, mostly, neglected fungal biodiversity and its critical role in this respect. Therefore, there is a great need for investigating the diverse fungal communities for new fungal strains, having new roles, instead of the already available ones.

This study was carried out to draw attention to the possibility of domiciliating a new beneficial microbiome into the plant tissue. In this connection, a pioneer isolate of *T. asperellum* ZNW, an endophytic fungus isolated from pea seeds, was domiciliated into the pea plant, representing a case study for the settlement of the bioagent fungus into plant tissue.

*Trichoderma* spp. is a diverse group of wonderful bioagents widely used in sustainable agriculture owing to their ability to enhance plant growth and productivity, and further alleviate abiotic and biotic stresses [37]. *Trichoderma* spp. are used in various remedies (foliar application, seed treatment, and/or soil remedy) for the management of diseases caused by bacteria, fungi, and nematodes [38].

Cultural, morphological, and microscopic (light, and SEM) identification of fungal isolate ZNW declares that the current isolate ZNW resembles the morphology of *T. asperellum* RMCK01 [39]. The morphology of *T. asperellum* ZNW growing on a PDA plate did not have the same morphology as observed in previous investigations, suggesting a new variant of *T. asperellum*. The conidia of *T. asperellum* ZNW are slightly globose in shape and the conidiophore is branched, which resembles *T. asperellum* Ta13 [40]. The morphological characters of *T. asperellum* ZNW have followed the classification of *T. asperellum* [18]. This data supports the claim that myco-diversity needs more studies to explore the undiscovered area of seed-borne fungi.

Identification of the damping-off phytopathogen, *G. ultimum* NZW, declared that its mycelia resemble those of *Pythium ultimum* var. *ultimum* associated with soybean damping-off [41]. SEM micrographs of *G. ultimum* NZW illustrated that its hyphae had a swelling end with a slightly globose shape, which was confirmed by other studies on *G. ultimum* strains [19,42,43]. All these works confirm the classification of the current fungus.

The pathogenicity of the current strain was confirmed by the pathogenicity test [44]. However, *Globisporangium ultimum* (Trow) (syn. *G. ultimum* Trow, syn. *G. ultimum Trow* var. *ultimum*) is an oomycete species that causes damping-off and/or root rot on a vast range of plants throughout the world [42,43,45].

The previous identification of both fungi was further confirmed. Molecular identification is frequently used due to its high specificity and sensitivity for the quick identification of numerous microorganisms [46]. The sequences of the tested fungi were related to similar strains in the GenBank. Both the bioagent and pathogenic fungi were identified as *T. asperellum* ZNW (OL604172) and *G. ultimum* NZW (MW922713), respectively, which came in line with the morphological identification features.

Molecular identification is highly specific and sensitive and is frequently used for fast identification. The nucleotide sequencing of the ITS region is similar in broad fungal classes, thus helping in revealing the interspecific and intraspecific differences among microorganisms. This region has non-functional sequences and is very variable among species, therefore, it can be accepted for the identification of very wide groups of fungal species. Technically, the multi-copy or repeated feature of the rDNA facilitates the amplification of the ITS regions from a tiny part of DNA, thus, providing fast and accurate identification compared to the other markers [47,48,49].

SEM investigation of the dual culture showed the ability of *T. asperellum* ZNW to mycoparasite and coil around *G. ultimum* NZW, causing damage, fractionation, and breakdown of *G. ultimum* NZW hyphae. Mycoparasitism is an effective method through which *Trichoderma* spp. physically contact and coil their hyphae around the hyphae of the pathogenic fungi, then the bioagent obtains nutrients from the dead host biomass after hydrolysis by the secretion of lytic enzymes and other metabolites [50,51].

However, there is no antagonistic activity of *T. asperellum* ZNW volatile organic compounds against *G. ultimum* NZW. This result means that *T. asperellum* ZNW may not produce volatile compounds or produce volatile compounds that did not inhibit *G. ultimum* NZW, the latter assumption may be the more accepted one. Conversely to our results, *T. asperellum* produced volatile compounds that had antifungal activity against other plant pathogens (*Curvularia aeria* and *Corynespora cassiicola*), enhanced the defense, and improved the growth of lettuce [52]. Moreover, *T. asperellum* T76-14 produced volatile organic compounds that inhibited *Fusarium incarnatum*, which causes fruit rot of muskmelons [53].

The biochemical feature of the bioagent was assessed as a possible role in the biological control process. Lytic activity of *T. asperellum* ZNW was searched on pea straw. This pea straw is hard to disintegrate due to its cellulose, hemicellulose, protein, and pectin content. Generally, such structures hinder fungal invasion into plant tissues. These enzymes are known as cell wall-degrading enzymes and play a role in the biocontrol process against the phytopathogen. Another suggested role is the facilitation of hyphal penetration to plant cell-wall tissue during the domiciliation process.

There was an obvious lytic activity of xylanase and cellulase that hydrolyzes hemicellulose (xylan) and cellulose, leading to the release of xylose and glucose units, respectively [47]. Whereas pectinase and proteases degrade pectin polymer and protein into galacturonic acid [47] and amino acids [28], respectively. The breakdown of plant tissues by these enzymes eases microbial entrance into plants [54], thus facilitating the domiciliation process.

Chitinase analyzes 1–4 β-glycoside linkage of N-acetyl-D-glucosamine in the fungal chitin; thus, acting as a bio-fungicide by antagonizing the other phytopathogens on host plants [55]. The growth medium was free from the chitin substrate, demonstrating that chitinase is produced constitutively by the *Trichoderma asperellum* ZNW.

Furthermore, *Trichoderma* spp. can biosynthesize many metabolites, which have antimicrobial activity against various phytopathogens, induce disease resistance, and enhance plant growth [56]. Besides many components, 9-octadecenoic acid (peak area-36.32%) was the main secondary metabolite detected in the filtrate of *T*. *asperellum* ZNW by GC-MS analysis. This component is presented in different *Trichoderma* strains in other investigations at lower concentrations in *T*. *asperellum* (peak area-1.57%) [57] and *T. hamatum* (peak area-1.77%) that inhibited *Candida albicans* [58]. The current compound, 9-octadecenoic acid, was reported to be belonging to the unsaturated fatty acids that have antimicrobial activity [59] and plays an obvious task in the plant’s defense against biotic and abiotic stresses [60]. Another fatty acid (9-hexadecenoic acid or palmitic acid) was present in the filtrate (peak area-3.96%) of this investigation, which was previously detected in the filtrate of *T. hamatum* [58] with antimicrobial activity [61]. Moreover, palmitic acid is found in the filtrate of *T*. *asperellum* and is also used for disease management and as a plant nutrient [62]. The possible role of palmitic acid is the ability to alter the rhizosphere microbial structure, control fungal disease, and encourage plant growth [63].

Oleamide, a TMS derivative (peak area-29.89%) present in the filtrate of *T. asperellum* ZNW, was previously reported as an amide derivative of the fatty acid oleic acid [64]. Fatty acids and their derivatives have vital roles in plant defense against bacterial and fungal pathogens [65]. Also, the detected 1,2-15,16-diepoxyhexadecane (peak area-8.13%) in this study may have antioxidant, anti-cancer, and anti-inflammatory activities [66]. Moreover, 1-Monopalmitin, 2TMS derivative (peak area-5.74%) detected in the filtrate of *T*. *asperellum* ZNW is a secondary metabolite in most active actinomycetes, having anticancer and antimicrobial activities [67]. In general, the antimicrobial effect of *T. asperellum* ZNW may be due to the synergism among the diverse chemicals, especially the main components, however, the synergistic effects between the main and the minor components exceed the expected individual outcome of each component alone [54,68].

Under greenhouse conditions, pea seeds were treated by the bioagent through biopriming with *T*. *asperellum* ZNW. Beneficial microbial inoculants are a vital method for improving the germination of seeds and/or enhancing growth, as well as the resistance of plants against several abiotic and biotic stresses. Moreover, seed biopriming permits the microorganisms, such as *Trichoderma* spp., to adhere to and enter the tissues [69,70,71].

Seed biopriming in *T. asperellum* ZNW increased the survival percentage and decreased percentages of rotted seeds and infected seedlings in the absence or presence of *G. ultimum*. The promotion of growth may be due to the action of the metabolites and the ability of fungal inoculation to reduce the disease (root rot and damping-off) and also to enhance the survival percentage of the plant [72]. The use of *T. asperellum* for biopriming of chili seeds reduced disease severity caused by *Fusarium solani* and enhanced seedling health [70].

*T. asperellum* ZNW enhanced the vegetative growth of pea plants that were planted in infested or non-infested soil due to its ability to promote plant growth, antagonize *G. ultimum* NZW, and its ability to induce the defense-related enzymes, and improve the concentration of photosynthetic pigment. These improvements resemble those reported on the growth parameter of tomato due to *T. asperellum*, which is used as a bioagent against damping-off disease [73].

Ultrastructural alterations were noticed by SEM in the cross sections of the root of the pea plant. *T*. *asperellum* ZNW could domiciliate and grow as an endophytic fungus in pea roots and lead to the thickened cell wall of vessels (xylem). Moreover, *T*. *asperellum* ZNW prevented *G. ultimum* NZW from invading the region of the root, in which *T. asperellum* ZNW is present. Furthermore, no traces of the pathogen could be observed in the region of *T. asperellum* ZNW. According to the authors’ information, this is the first trial that indicates the ability of domiciliation of *Trichoderma* spp. into plant tissue so that it can grow as an endophytic fungus. The growth of *T*. *asperellum* ZNW as an endophytic fungus gave a high opportunity for plant protection from phytopathogens and allow the plant to get several profits from *T*. *asperellum* ZNW metabolites. *T*. *asperellum* ZNW led to an increase in thickness of the xylem of pea roots, this may be due to the lignification, i.e., lignin accumulation in the cell wall of the root that is accompanied by low methyl-esterified pectin in intercellular spaces, which increases in cell wall thickness improved wheat protection against pathogens [74].

Additionally, the enzymatic system of *T. asperellum* ZNW enables partial maceration of plant tissue that allows the smooth penetration, and domiciliation of *T. asperellum* ZNW into plant tissue without causing any pathogenic symptoms. Generally, *Trichoderma* spp. do not have any virulence factor and, consequently, no disease development occurs.

*T. asperellum* ZNW enhanced the induced resistance and increased the production of total phenols and improved the activities of peroxidase and polyphenol oxidase in pea planted. This, in turn, leads to an increase in defense response against viral, bacterial, and fungal pathogens [7,8,9]. Phenolic compounds have significant roles in plant health through increasing colonization of beneficial microorganisms and inhibition of insect attack and/or pathogen infection [75]. Phenolic compounds have antioxidant and antimicrobial properties and protect the plant tissues from poisonous effects, resulting from reactive oxygen species [76]. Under adverse conditions, phenolics accumulate in the plant tissues and play a vital role in the regulation of different environmental stresses, such as nutrient deficiency, low temperatures, and high light [77]. Peroxidase and polyphenol oxidase are defense-related enzymes that enhance the defense against phytopathogens infections and improve plant resistance against biotic and abiotic stress [78].

*T. asperellum* ZNW enhanced chlorophylls and carotenoid content in pea plants. *T. asperellum* could enhance the pea plant’s health and enhance the activity of the photosynthesis process [79]. The improvement of chlorophyll concentration refers to the enhancement of the plant’s defense against abiotic and biotic stresses [80]. *T. asperellum* enhanced chlorophylls and carotenoids content when used as biocontrol of plant pathogens [81]. Furthermore, *T. asperellum* enhanced the concentration of photosynthetic pigments and helped to overcome drought stress [82].

The changes in the biological features in the rhizosphere soil of pea were explored. The activity of the soil dehydrogenase was enhanced due to the application of *T. asperellum*. *Trichoderma* spp. was reported to improve the activity of dehydrogenase and positively amend the soil properties [83]. Dehydrogenase is one of the most important soil enzymes, which is an indicator of soil quality due to different stress, or management practices. Enzyme activity in the soil can be changed as a result of the activities of the soil’s microbes [84]. Since dehydrogenases are present intracellularly in the living microbial cells and do not accumulate out of the microbial cells, it is considered an indicator of the microbial activity in the soil; furthermore, dehydrogenases play a vital role in the biological oxidation of the organic matter in the soil by transferring hydrogen from organic substrates to inorganic acceptors [85]. The enhancement of dehydrogenase is a good sign of the improvement of microbial population and physicochemical soil properties [86,87].

*T. asperellum* ZNW led to an alteration in the bacterial and fungal communities in the rhizosphere, where both microbial counts decreased greater after 90 days than after 60 days. This effect of *T. asperellum* ZNW on the rhizosphere microbial count may be directly and/or indirectly related to plant physiology, and root exudates in soil. This effect may be due to the competition of *Trichoderma* with other microbes, leading to an alteration in the population of the soil microbiome. Also, the activity of soil enzymes may have a vital role in changing the microbial community [88]. The relative abundance of *Trichoderma* spp. in the soil causes increases in the availability of soil nutrients, and changes the rhizosphere chemical structure, thus altering soil microbial communities [10]. *T. asperellum* causes an increase in the total count of beneficial microbes and a decrease in the total count of phytopathogens in the plants’ rhizosphere that help control plant diseases and promote growth [89]. *Trichoderma* spp. can boost the physiology of plants and can modulate root exudates that cause a change in nature and kind of the plant-microbe interaction [90]. Root exudates are secretions produced by roots that are considered a basis of nutrition for beneficial soil microbial communities [91]. Root exudates enhance soil respiration and positively change the configuration of the microbial community, which effectively enhances the growth of plants [92]. The root exudates decrease near the end of plant age leading to decreasing in the microbial count of rhizosphere microbial communities [93].

The biopriming of pea seeds in *T. asperellum* ZNW led to an increase in the productivity of pea plants and yield parameters. This is an expected outcome due to the enhancement of plant growth and physiology, as well as the management of phytopathogen, leading to an enhancement in induced resistance and also an enhancement of the rhizosphere microbial community. *T. asperellum* could promote different plants and control diseases, leading to an increase in yield, as occurs when controlling late-wilt disease, promoting growth, and increasing the yield of maize [94]. Furthermore, *T. asperellum* could reduce *Pythium* root rot, promote growth, and improve the productivity of lettuce [95].

## 5. Conclusions

Summing up, this contemporary report offers background knowledge on the possibility of the domiciliation of a pioneer isolate of *T. asperellum* ZNW into pea plant tissue as a new habitation for the bioagent fungus. The biochemical features of *T. asperellum* ZNW were evaluated in vitro before being evaluated under in vivo conditions, and SEM examination. However, this investigation is the first record of the pathogenicity of *G. ultimum*, as a damping-off causative agent in pea. *T. asperellum* ZNW could inhibit *G. ultimum* NZW growth through mycoparasitism and producing hydrolytic enzymes (xylanase, cellulase, pectinase, protease, and chitinase) and different bioactive metabolites. *T. asperellum* ZNW can domiciliate the root of the pea and reduce rotted seeds and infected seedlings, and also improve survival. Additionally, the study confirms the positive effect of the domiciliated *T. asperellum* ZNW, under infection, on plant health. *T. asperellum* ZNW improved the disease resistance, photosynthetic pigmentation, vegetative growth, and yield of pea. This investigation is the first attempt to apply the simple seed biopriming technique to domiciliate *T. asperellum* ZNW in pea tissue as an endophytic fungus. Nevertheless, our results may enable a deep understanding of plant-fungal interaction. So, it would be worthwhile to perform further evaluation, and more studies are encouraged, to extend the application of the current data on other beneficial microorganisms on other economic plants.

## Figures and Tables

**Figure 1 microorganisms-11-00198-f001:**
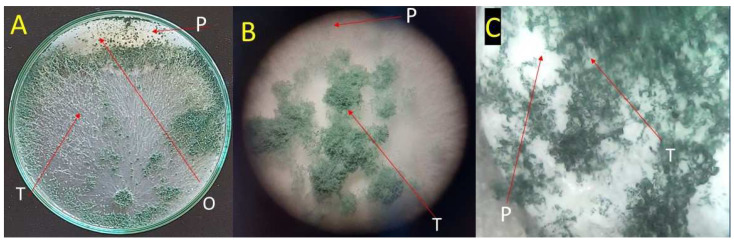
*Trichoderma* spp. antagonizes (overgrows) the cause of damping-off fungus of pea on PDA plates; (**A**): the two fungi growing on the same PDA plate, showing the overgrowth; (**B**,**C**): overgrowth view of *Trichoderma* spp. on the fungal pathogen at 10×, and 100×, respectively. Where T-arrow: *Trichoderma* spp., P-arrow: fungal pathogen, and O-arrow: overgrowth region.

**Figure 2 microorganisms-11-00198-f002:**
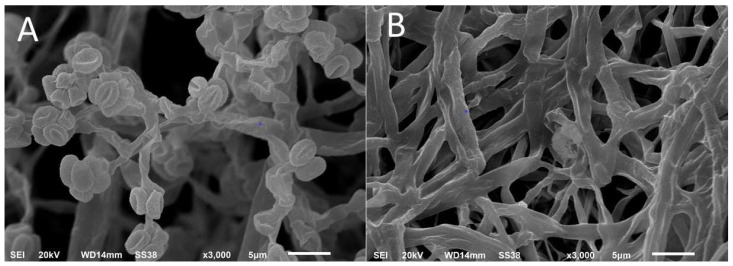
Scanning electron microscopy of the two investigated fungi; (**A**): *Trichoderma asperellum* ZNW, (**B**): *Globisporangium ultimum* NZW.

**Figure 3 microorganisms-11-00198-f003:**
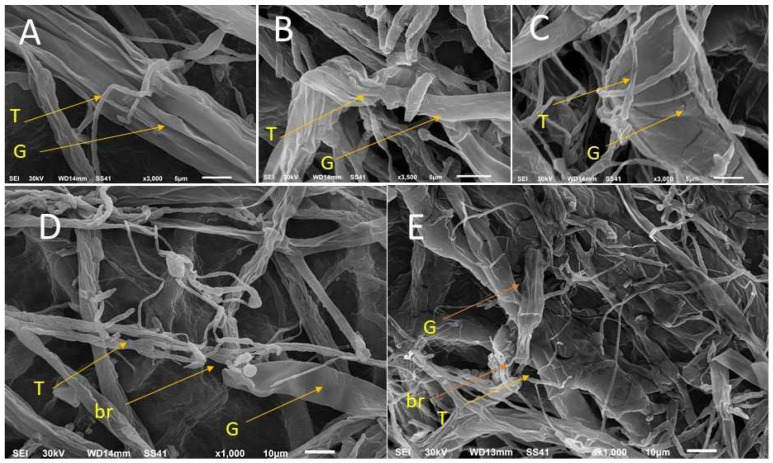
Mycoparasitism of *Trichoderma asperellum* ZNW on *Globisporangium ultimum* NZW (after their first contact in a dual culture) under the scanning electron microscope. (**A**): overgrowth of *T. asperellum* ZNW hyphae (T-arrow) on *G. ultimum* NZW hyphae (G-arrow), (**B**): coiling of *T. asperellum* ZNW hyphae around *G. ultimum* NZW hyphae, (**C**): continuous coiling, (**D**): the breakdown (br arrow) of *G. ultimum* NZW hyphae, and (**E**): the breakdown and collapse of *G. ultimum* NZW hyphae.

**Figure 4 microorganisms-11-00198-f004:**
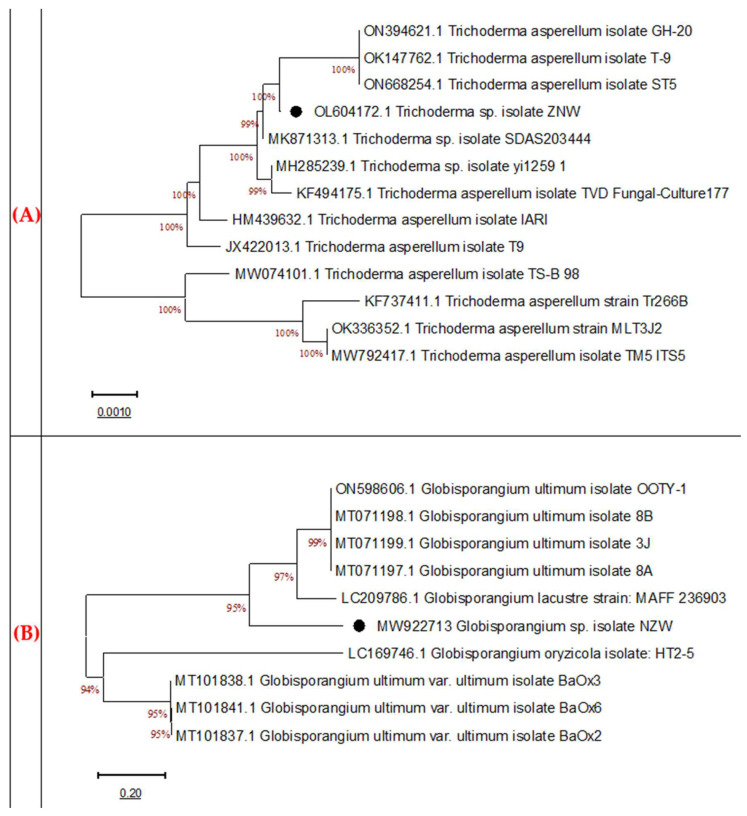
Molecular phylogenetic tree of the ITS for *Trichoderma asperellum* ZNW (**A**) and *Globisporangium ultimum* NZW (**B**) relation to the sequences obtained from GenBank.

**Figure 5 microorganisms-11-00198-f005:**
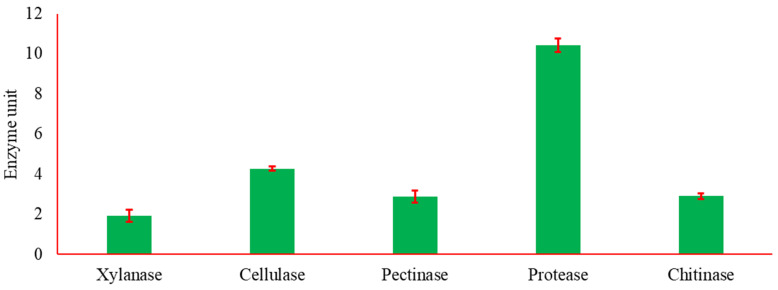
The hydrolytic enzymes’ activity of the bioagent *T. asperellum* ZNW (n = 3).

**Figure 6 microorganisms-11-00198-f006:**
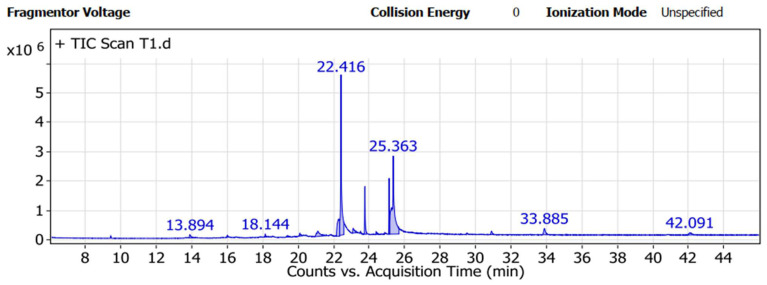
GC-MS of filtrate of *Trichoderma asperellum* ZNW.

**Figure 7 microorganisms-11-00198-f007:**
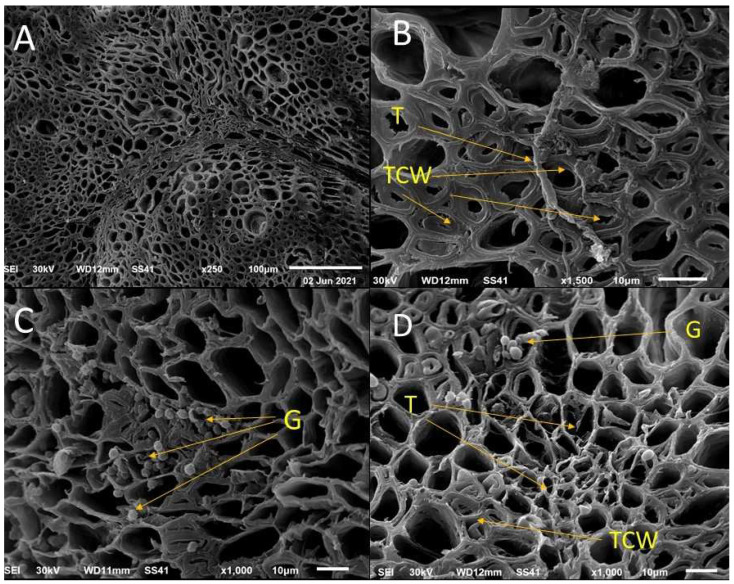
Micrographs of the SEM cross-section of pea root after 30 days of planting showing; (**A**): normal control plants, (**B**): plants treated with *T. asperellum* ZNW only, (**C**): plants grown in infested soil by *G. ultimum* NZW, and (**D**): plants treated with *T. asperellum* ZNW under infested soil by *G. ultimum* NZW. T-arrow: *T. asperellum* hyphae, TCW-arrow: thickening in cell wall vessels (TCW), G-arrow; *G. ultimum* NZW spores.

**Table 1 microorganisms-11-00198-t001:** Metabolites present in *Trichoderma asperellum* ZNW filtrate as indicated by GC-MS analysis.

Peak	RT	Name	Formula	Area	Area Sum %
1	13.894	Butylated Hydroxytoluene	C_15_H_24_O	727,422.65	1.16
2	16.011	Decane, 2,3,5,8-tetramethyl-	C_14_H_30_	305,789.06	0.49
3	18.144	Tetradecane, 2,6,10-trimethyl-	C_17_H_36_	260,402.83	0.41
4	19.417	Dodecanoic acid, 3-hydroxy-	C_12_H_24_O_3_	380,111.62	0.6
5	20.096	7-Hexadecenal, (Z)-	C_16_H_30_O	472,179	0.75
6	21.113	9-Hexadecenoic acid	C1_6_H_30_O_2_	2,494,012.6	3.96
7	22.273	1,2-15,16-Diepoxyhexadecane	C_16_H_30_O_2_	5,117,326.6	8.13
8	22.416	Oleamide, TMS derivative	C_21_H_43_NOSi	18,817,525	29.89
9	23.102	9,12-Octadecadienoyl chloride, (Z,Z)-	C_18_H_31_ClO	1,142,391.4	1.81
10	23.765	1-Monopalmitin, 2TMS derivative	C_25_H_54_O4Si_2_	3,615,270.1	5.74
11	24.398	12-Methyl-E,E-2,13-octadecadien-1-ol	C_19_H_36_O	403,516.61	0.64
12	25.129	Glycerol monostearate, 2TMS derivative	C_27_H_58_O_4_Si_2_	3,318,708	5.27
13	25.363	9-Octadecenoic acid (Z)-	C_18_H_34_O_2_	22,870,741	36.32
14	30.901	1,4-Benzenediol, 2,6-bis(1,1-dimethylethyl)-	C_14_H_22_O_2_	613,236	0.97
15	33.885	Acetic acid, (1,2,3,4,5,6,7,8-octahydro-3,8,8-trimethylnaphth-2-yl)methyl ester	C_16_H_26_O_2_	1,571,281.1	2.5
16	42.091	1,4-Bis(trimethylsilyl)benzene	C_12_H_22_Si_2_	856,462.94	1.36

**Table 2 microorganisms-11-00198-t002:** Disease assessment of pea damping-off under inoculation with *Trichoderma asperellum* ZNW and/or *G. ultimum* NZW.

Treatment	Rotted Seeds, %	Infected Seedling, %	Survival, %
Fungicide	10.00 ± 2.00 b	10.00 ± 2.00 a	80.00 ± 5.00 a
Control	13.33 ± 2.89 b	16.67 ± 1.15 a	70.00 ± 4.01 ab
*T. asperellum*	10.00 ± 2.00 b	10.00 ± 2.22 a	80.00 ± 2.03 a
Fungicide + *G. ultimum*	20.00 ± 2.22 b	10.00 ± 2.22 a	70.00 ± 2.58 ab
*G. ultimum*	46.67 ± 2.89 a	18.33 ± 3.51 a	35.00 ± 2.43 b
*T. asperellum* + *G. ultimum*	11.67 ± 2.89 b	10.00 ± 1.20 a	78.33 ± 2.89 a

In each column, numbers that share the same letter(s) are not significantly different (*p* ≤ 0.05, n = 5).

**Table 3 microorganisms-11-00198-t003:** Vegetative growth parameters of pea under the action of bioagent and the pathogen.

Treatment	Root Length (cm)	Shoot Length (cm)	Plant Height (cm)	Root Fresh Weight (g)	Root Dry Weight (g)	Plant Fresh Weight (g)	Plant Dry Weight (g)	Leaf Area (cm^2^)
Fungicide	8.00 ± 1.00 ab	24.00 ± 2.11 a	32.00 ± 1.73 a	0.280 ± 0.026 ab	0.047 ± 0.006 ab	7.24 ± 0.74 a	1.307 ± 0.064 a	99.4 ± 6.3 b
Control	7.67 ± 0.58 b	25.33 ± 1.15 a	33.00 ± 1.00 a	0.287 ± 0.025 a	0.083 ± 0.015 a	5.57 ± 0.23 a	1.037 ± 0.093 ab	130. 8 ± 5.5 a
*T. asperellum*	8.00 ± 1.00 ab	22.67 ± 1.15 a	30.67 ± 1.55 a	0.223 ± 0.031 ab	0.070 ± 0.010 a	5.35 ± 0.64 a	0.970 ± 0.161 ab	107.3 ± 6.8 ab
Fungicide + *G. ultimum*	7.67 ± 1.15 b	21.67 ± 0.58 a	29.33 ± 1.53 a	0.183 ± 0.035 ab	0.087 ± 0.011 a	5.52 ± 0.65 a	0.953 ± 0.076 ab	96.6 ± 4.3 b
*G. ultimum*	5.33 ± 0.58 c	8.67 ± 1.53 b	14.00 ± 2.75 b	0.080 ± 0.010 b	0.020 ± 0.010 b	1.59 ± 0.13 a	0.230 ± 0.036 b	31.4 ± 3.8 c
*T. asperellum* + *G. ultimum*	10.00 ± 1.00 a	22.67 ± 1.55 a	32.67 ± 1.55 a	0.177 ± 0.015 ab	0.060 ± 0.010 ab	5.71 ± 0.51 a	1.213 ± 0.042 a	104.7 ± 4.8 ab

In each column, numbers that share the same letter(s) are not significantly different (*p* ≤ 0.05, n = 5).

**Table 4 microorganisms-11-00198-t004:** Defense-related enzymes and total phenols of pea plants under inoculation with *T. asperellum* ZNW and/or *G. ultimum* NZW.

Treatment	Peroxidase (U)	Polyphenol Oxidase (U)	Total Phenol (mg g^−1^ Fresh Weight)
Fungicide	7.3 ± 1.2 bcd	10.0 ± 2.2 ab	69.8 ± 2.1 ab
Control	6.0 ± 1.3 cd	8.0 ± 2.2 ab	66.9 ± 1.8 ab
*T. asperellum*	10.0 ± 1.8 abc	11.3 ± 1.2 a	49.9 ± 1.2 c
Fungicide + *G. ultimum*	12.0 ± 2.0 ab	7.3 ± 1.2 bc	62.2 ± 2.4 b
*G. ultimum*	4.7 ± 1.2 d	4.0 ± 0.3 c	43.0 ± 1.6 c
*T. asperellum* + *G. ultimum*	14.0 ± 2.2 a	11.3 ± 1.2 a	71.0 ± 2.2 a

In each column, numbers that share the same letter(s) are not significantly different (*p* ≤ 0.05, n = 3).

**Table 5 microorganisms-11-00198-t005:** Chlorophylls and carotenoids content (mg g^−1^ fresh weight) of pea plants under inoculation with *T. asperellum* ZNW and/or *G. ultimum* NZW.

Treatment	Chlorophyll a	Chlorophyll b	Total Chlorophylls	Carotenoids
Fungicide	1.642 ± 0.036 d	0.517 ± 0.033 cd	2.159 ± 0.069 e	0.520 ± 0.045 a
Control	2.010 ± 0.033 c	0.590 ± 0.019 bc	2.600 ± 0.107 cd	0.457 ± 0.056 a
*T. asperellum*	2.073 ± 0.014 c	0.368 ± 0.041 d	2.351 ± 0.179 de	0.440 ± 0.021 a
Fungicide + *G. ultimum*	2.049 ± 0.011 c	0.764 ± 0.027 b	2.813 ± 0.017 c	0.300 ± 0.012 a
*G. ultimum*	2.476 ± 0.015 b	0.683 ± 0.021 bc	3.159 ± 0.040 b	0.469 ± 0.009 a
*T. asperellum* + *G. ultimum*	2.707 ± 0.066 a	0.987 ± 0.052 a	3.694 ± 0.059 a	0.411 ± 0.025 a

In each column, numbers that share the same letter(s) are not significantly different (*p* ≤ 0.05, n = 3).

**Table 6 microorganisms-11-00198-t006:** Evaluation of *T. asperellum* ZNW on the yield of pea under infection with *G. ultimum*.

Treatment	Pods Number	Pods Fresh Weight (g)	Pods Dry Weight (g)	Weight of 100 Seeds (g)
Fungicide	61.00 ± 2.25 b	61.43 ± 2.16 c	10.34 ± 0.81 b	5.38 ± 0.60 d
Control	27.00 ± 1.13 f	58.61 ± 1.52 d	6.12 ± 0.65 c	6.15 ± 0.75 c
*T. asperellum*	52.33 ± 1.53 c	78.27 ± 2.20 b	9.36 ± 0.84 b	6.23 ± 0.65 c
Fungicide + *G. ultimum*	44.67± 1.25 d	36.30 ± 1.36 e	13.23 ± 0.55 b	7.77 ± 0.35 b
*G. ultimum*	35.00 ± 1.65 e	28.65 ± 1.25 f	5.05 ± 0.50 c	6.23 ± 0.45 c
*T. asperellum* + *G. ultimum*	70.67 ± 2.55 a	81.07± 2.15 a	14.28 ± 0.76 a	8.63 ± 0.85 a

In each column, numbers that share the same letter(s) are not significantly different (*p* ≤ 0.05, n = 5).

**Table 7 microorganisms-11-00198-t007:** The activity of dehydrogenase enzyme estimated in the rhizosphere soil of pea plants.

Treatment	Dehydrogenase Activity (µg TPF g^−1^ Dry Soil min^−1^)
Fungicide	0.283 ± 0.004 b
Control	0.213 ± 0.002 d
*T. asperellum*	0.242 ± 0.003 c
Fungicide + *G. ultimum*	0.246 ± 0.004 c
*G. ultimum*	0.211 ± 0.002 d
*T. asperellum* + *G. ultimum*	0.404 ± 0.003 a

In each column, numbers that share the same letter(s) are not significantly different (*p* ≤ 0.05, n = 3).

**Table 8 microorganisms-11-00198-t008:** Total bacterial and fungal count in the rhizosphere soil of pea plants after 60 and 90 days of planting.

Treatment	**Microbial Count (log CFU g^−1^ Dry Soil)**			
	Bacteria		Fungi	
	60 Days	90 Days	60 Days	90 Days
Fungicide	6.72 ± 0.01 c	6.04 ± 0.04 b	4.84 ± 0.06 a	2.00 ± 0.10 c
Control	6.82 ± 0.01 b	6.29 ± 0.05 a	4.77± 0.07 ab	2.77 ± 0.07 a
*T. asperellum*	6.60 ± 0.02 d	6.20 ± 0.05 a	4.99 ± 0.04 a	2.59 ± 0.11 ab
Fungicide + *G. ultimum*	6.40 ± 0.04 f	6.01 ± 0.01 b	4.56 ± 0.07 bc	2.72 ± 0.05 ab
*G. ultimum*	6.99 ± 0.04 a	5.64 ± 0.04 c	4.82 ± 0.03 a	2.48 ± 0.31 b
*T. asperellum* + *G. ultimum*	6.48 ± 0.01 e	6.02 ± 0.06 b	4.46 ± 0.02 c	2.10 ± 0.17 c

In each column, numbers that share the same letter(s) are not significantly different (*p* ≤ 0.05, n = 3).

## Data Availability

All data generated and/or analyzed during this study are included in this published article.

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
