# Peer review of "Domiciliation of Trichoderma asperellum Suppresses Globiosporangium ultimum and Promotes Pea Growth, Ultrastructure, and Metabolic Features"

_microorganisms, 2023, doi:10.3390/microorganisms11010198_

Round 1
Reviewer 1 Report
The research was interesting to me. However, it was disappointing that many of the manuscripts were not written correctly.
I thought the Article Title did not represent the content correctly.
There was no record of how many replication experiments were performed in analyzes such as enzyme activity.
Although significant differences were shown in Tables 2-8, the standard deviations of each value were not shown, so it was not possible to judge whether the statistical analysis had been performed correctly.
In the manuscript, there were any sentences with a period in the wrong place.
There were too long sentences to understand (especially the discussion section).
There was an error in the format of the references (line 671).
There were many other mistakes, so please check the content thoroughly before considering submission.
Author Response
We would like to thank you for your constructive comments concerning our Manuscript. These comments are all valuable and helpful for improving our manuscript. All the authors have seriously discussed all these comments. We have tried our best to modify the manuscript to meet the requirements. in the revised version. Point-by-point responses to the comments are listed below.

Reviewer 2 Report
The manuscript contains quite interesting research results. Needs some refinement.
Comments
Before the "." no spaces are put.
References
Please remove publications older than 10 years (16, 17, 25, 26, 29, 31, 44, 59, 65) and above all from the last century (18, 20, 21, 22, 24, 27, 33, 34 35)
Author Response

(The authors gave the same response as above.)

Round 2
Reviewer 1 Report
I've verified that the things I pointed out have been properly fixed.
Since the manuscript has been proofread, please change it to the correct manuscript if it is to be published in the journal.